

# Monitoring the global ocean heat content from space geodetic observations to estimate the Earth energy imbalance

Florence Marti[1], Victor Rousseau[1], Michaël Ablain[1], Robin Fraudeau[1], Benoit Meyssignac[2], Alejandro Blazquez[2]

[1]Magellium, Ramonville Saint Agne, 31520, France
[2]Université de Toulouse, LEGOS, Toulouse, 31400, France

*Correspondence to*: Florence Marti (florence.marti@magellium.fr)

**Abstract.** An improved spatial geodetic approach is presented for estimating the global ocean heat content (GOHC) change and the Earth energy imbalance (EEI) over 1993-2022. The geodetic estimate of the EEI shows a significant positive trend of 0.75 W m$^{-2}$ over the period 1993-2022, indicating accelerated warming of the ocean and increasing EEI, in line with CERES observations. Comparisons with in situ data GOHC changes shows good agreement over 2005-2019. This study highlights the importance of rigorously estimating uncertainties based on space geodetic data to robustly assess EEI changes.

## 1 Introduction

The ocean absorbs much of the excess energy stored by the Earth system that results from the greenhouse gas emission by human activities in the form of heat (~91%; Church et al., 2011; Levitus et al., 2012; Meyssignac et al., 2019; von Schuckmann et al., 2020, 2022; Foster et al., 2021). As the ocean acts as a huge heat reservoir, global ocean heat content (GOHC) is therefore a key component in the Earth's energy budget. An accurate knowledge of the GOHC change allows us to assess the Earth energy imbalance (EEI), which refers to the difference between the amount of energy the Earth receives from the sun and the amount of energy it radiates back into space. A community effort (Meyssignac et al., 2019) depicted the various methodologies to estimate EEI from the GOHC, including the use of temperature and salinity profiles (von Schuckmann et al., 2020, 2022), the measurement of the ocean thermal expansion from space geodesy (Marti et al., 2022), ocean reanalysis (Stammer et al., 2016), and net flux measurements (Kato et al., 2018; L'Ecuyer et al., 2015), L'Ecuyer et al. 2015). Among these approaches, the space geodetic approach, detailed in Marti et al. (2022), leverages the maturity of satellite altimetry and gravimetry measurements, enabling precise, extensive spatial and temporal coverage, and full-depth estimates of ocean thermal expansion. As the EEI magnitude is small (0.5-1.0 W m$^{-2}$, von Schuckmann et al., 2016) compared to the amount of energy entering and leaving the climate system (~340 W m$^{-2}$, L'Ecuyer et al. 2015), a high level of precision and accuracy are required to estimate the EEI mean (< 0.3 W m$^{-2}$) and its time variations at decadal scale (< 0.1 W m$^{-2}$ ; Meyssignac et al., 2019). In this regard, the space geodetic approach emerges as a promising candidate capable of meeting the stringent EEI precision and accuracy requirements (Meyssignac et al., 2019; Marti et al., 2022).



In this study, our primary objective is to present the updated space geodetic GOHC and EEI estimates and the improvement since Marti et al. (2022), including several major evolutions in the input data, algorithms and a temporal extension into the past since 1993. The secondary objective is to compare this updated space geodetic monthly GOHC product with GOHC time series derived from in situ observations. To ensure a consistent and homogeneous treatment, we apply the same processing method to estimate the EEI from the different yearly GOHC time series considered. The obtained EEI estimates

are then compared to the net flux at the top of atmosphere (TOA) derived from the Clouds and the Earth's Radiant Energy System (CERES) mission, which serves as a reference for EEI time variations.

## 2 Data and method

The space geodetic approach consists in deriving the ocean heat content change from the steric sea level change (i.e. the ocean expansion)  inferred by satellite observations. We present here an update of the technique for estimating the GOHC

change and the EEI, which relies on the existing work (Marti et al., 2022) but also benefits from the progress made more recently at regional scales (Rousseau et al., under revision).

Space geodetic observations are consistent with the ones used in Marti et al. (2022). The total sea level change is derived from altimetry sea-level gridded products data from the Copernicus Change Climate Change service (C3S) [1]. A correction for TOPEX-A drift is applied (Ablain et al., 2017) as well as a correction for the Jason-3 radiometer drift (Barnoud et al.,

2023). The manometric sea level change is estimated from an update of Blazquez et al. (2018) gravimetric solution ensemble (V1.6) [2]. In this update, we identified a sub-sample of the ensemble which relies on a single geocenter correction based on Sun et al. (2016) and whose mean is used as our best estimate.

The GOHC change is obtained as the sum of regional ocean heat content (OHC) estimated on a 1°x1° grid in accordance with Rousseau et al. (under revision). The uncertainties, their characterisation and their propagation from the input data until

the GOHC change and EEI are made at global scale in a similar manner to Marti et al. (2022).

Based on the sea level equation, space geodetic data allow estimating the steric sea level (SSL) change. We derive GOHC change from the SSL change neglecting the effect of the halosteric sea level change (HSL) because the impact of salinity changes on SSL is very small at global scale. Moreover, recent studies highlight that in situ salinity datasets from Argo floats - from which we are able to derive HSL change - present an instrumental drift since 2016 due to anomalies on part of the

conductivity sensors (Wong et al., 2020), leading to a significant drift in the global mean HSL estimates from 2016 onwards (Barnoud et al. (2021)).

The OHC change is then obtained from the ratio of the SSL change and the integrated expansion efficiency of heat (IEEH) coefficient. The IEEH is computed at regional scale (1°x1°) from temperature and salinity data from the ECCO ocean reanalysis [3]. In previous versions, the IEEH was computed at global scale (Marti et al., 2022) and regional scales

(Rousseau et al., under review)  from in situ temperature/salinity profiles (mainly Argo floats). The two advantages of relying on ECCO here is to extend the spatial area over which the IEEH is computed including now the coasts and the high



latitudes, and to take into account the deep ocean down to 6000m. In this paper, we have made the approximation that the IEEH is constant over time. This is justified at global scale because the heat pattern of the ocean does not change significantly on decadal time scales (Kuhlbrodt and Gregory, 2012).

Argo-derived global IEEH ranges from 1.45 10-1 m YJ$^{-1}$ for a depth down to 2000 m to 1.67 10-1 m YJ$^{-1}$ for a depth down to 6000 m. Using the ECCO ocean reanalysis [3] instead of Argo data provides very similar global IEEH values (see Table 1). The ECCO reanalysis allows to get an estimate of the global IEEH down to the bottom of the ocean and close to the coast. Over the entire ocean the ECCO reanalysis indicates an IEEH of 1.50 10-1 m YJ$^{-1}$. The global IEEH uncertainty of 1 10-3 m YJ$^{-1}$ ([5%,95%] confidence interval level) is obtained by considering the spread in the Argo-derived global IEEH estimates

over the Argo mask (Marti et al., 2022).

In this study we propose a temporal extension of the geodetic estimate of GOHC and EEI into the past from January 1993 (at the beginning of precise satellite altimetry) onwards. As space gravimetry observations are not available before 2002 (the GRACE mission was launched in March 2002), the global mean sea level barystatic component is extended into the past with the sum of the individual contributions to manometric sea level from Greenland, Antarctica, mountain glaciers and from

terrestrial water storage. These different contributions are derived from the SLBC_cci product [4].

After calculating the GOHC, the EEI is then obtained from the time derivative of the GOHC - by applying a central finite difference scheme - and accounting for the heat fraction that is entering the ocean (which is 91%) - the rest being captured by the atmosphere, land and cryosphere (Forster et al., 2021). As described in Marti et al. (2022), the OHC change needs to be filtered out beforehand by applying a Lanczos low-pass filter at 3 years to remove signals related to ocean-atmosphere

exchanges which does not correspond to any response to global warming (Palmer and McNeall, 2014) and must therefore be removed to infer EEI variations. However, unlike Marti et al. (2022), we applied this temporal filter to regional spatial scales before summing the regional OHC estimates to obtain the GOHC. The following equation summarises how the EEI is derived from GOHC:

$$EEI(t) = \frac{dGOHC_{filtered,adjusted}(t)}{dt} \times \frac{1}{\alpha}, with \, \alpha = 0.91 \, , \qquad (1)$$

It is worth noting that the impact of performing the filtering step at regional scales rather than global scale is low on the GOHC estimate, but much more significant on the EEI estimate. It is because the filtering step allows to filter out the noise before the calculation of the time derivative and thus it minimises the noise amplification in EEI induced by the time derivation.

In order to assess the GOHC and EEI estimates, the estimation of their uncertainties is a key point. Briefly, the method

developed (described in Marti et al., 2022) consists in calculating the error variance-covariance matrices of the global mean sea level (GMSL) change data record and the barystatic sea level data record and then propagating these error variance-covariance matrices to the GOHC and the EEI estimates. The characterisation of uncertainties is similar to that used by Marti et al. (2022). For the GMSL uncertainties, we have used an updated altimetry uncertainty budget provided by Guérou et al.



(2022), mainly extended over the Jason-3 period (until 2021). For the barystatic sea level uncertainties, we have calculated
the dispersion of the gravimetry ensemble [2]. Note that this uncertainty is not centred on the barystatic best estimate (see
Figure 1). Besides, an uncertainty on the heat fraction entering the ocean has been introduced ([89%, 93%]) to account for
the different estimates from the literature (e.g. (Church et al., 2011; Levitus et al., 2012; von Schuckmann et al., 2020;
Forster et al., 2021; von Schuckmann et al., 2023). From the covariance matrices, we are able to obtain the uncertainty
associated with the means, trends or accelerations at any time scales based on an ordinary least square regression.

The space geodetic GOHC and EEI estimates [5] have been compared to other estimates mostly based on in situ data. We
analyse the geodetic estimate to 3 ocean monitoring indicators (OMIs) delivered by CMEMS [6] and based on in situ
observations (CORA, ARMOR-3D, and CORA processed by von Schuckmann and Le Traon (2011) (later "CORA-2011").
Note that ARMOR-3D also use space measurements (altimetry and sea surface salinity and temperature) in addition to in situ
observations to derive a GOHC estimate. The OMIs have been amended with a deep ocean warming estimate of +0.068 W
m$^{-2}$ from (Purkey and Johnson, 2010) to encompass the entire water column and account for the deep ocean's substantial
thermal influence below 2000 m. The CORA-2011 dataset is delivered together with an uncertainty envelope whose
estimation is described in von Schuckmann and Le Traon (2011). We also compare the geodetic estimate of the GOHC to
the recent Global Climate Observing System (GCOS) ensemble [7] composed of 16 time series based on subsurface
temperature measurements and representative of the full water column. For the GCOS GOHC ensemble trend we use the
uncertainty indicated in von Schuckmann et al. (2023) for the period 2006-2020. Note that CORA and CORA-2011 time
series are included within the GCOS ensemble. In addition, we compare the geodetic GOHC estimate with GOHC estimates
derived from gridded fields of temperature and salinity products provided by 5 Argo centres, namely ISAS20 - IFREMER
[8], SIO (Scripps Institution of Oceanography) [9], EN4 using two sets of corrections (Cheng et al., 2014; Gouretski and
Cheng, 2020) [10], JAMSTEC version 2021 [11] and NOAA (National Oceanic and Atmospheric Administration) [12]
datasets. The Argo resulting GOHC change estimates have been extended with Purkey and Johnson (2010) deep ocean
contribution. It should be noted that both GCOS ensemble and OMIs are made up of yearly time series, whereas the space
geodetic GOHC estimates are monthly, which restricts comparisons to interannual scales. Comparisons are thus led on the
basis of annual time series, both for trend and variability study.

For the EEI comparison, each of the GOHC change time series mentioned above have been derived to obtain the EEI using
the same method: annual GOHC change data are linearly interpolated on a monthly time scale so the derivative is made on a
monthly time scale. The CERES Energy Balanced and Filled (EBAF) product [13] is used as a reference for the EEI
variability assessment because it is totally independent and it is known to reproduce precisely the EEI variations with
uncertainties of the order of a few tenth of W.m-2.

The data used for this study are described in Table 2, both for the calculation of GOHC and EEI estimates and for their
intercomparison.



## 3 Results

The space geodetic GOHC change (called LEGOS-Magellium) is plotted in Figure 1 from September 1993 to May 2022. It highlights a trend of +0.75 W m$^{-2}$ for the whole period, providing an estimate of the global ocean heat uptake (GOHU) and

indicating the rate of heat accumulation in the ocean. The uncertainty range for this GOHU is [0.61; 1.04] W m$^{-2}$ meaning the GOHU is significantly positive over 1993-2022. In the same figure, we also superimpose the GOHC change time series from GCOS.

The area covered by both datasets is not identical with differences in coastal areas (areas less than 100 km from the coast are excluded for spatial geodetic data, while a 300 m bathymetry criterion is applied for each GCOS ensemble member) and also

in latitudes (GCOS members are limited to the latitude 60° while the geodetic method goes up to 66°). As a result, GCOS solutions are derived from data spanning 76% of the total ocean surface, while the geodetic approach covers 87%. As OHC is an integrative variable, the GOHC change estimates are very sensitive to spatial coverage which may explain some differences in trend at global scale. Over their respective area of data availability, the trend of GCOS OHC ensemble is lower (0.60 [0.39; 0.82]), but still in agreement with the space geodetic within their confidence interval (0.73 [0.59; 1.02]). When

considering the same spatial extension as the GCOS ensemble, the space geodetic GOHC trend drops to 0.62 [0.50; 0.88] W m$^{-2}$ and is closer to that of the GCOS ensemble.

We compare the geodetic GOHC trends with all the other estimates (Figure 2) over the common period of availability 2005-2019. In a general manner the space-geodetic approach shows a more pronounced trend in GOHC than approaches based on in situ data (Hakuba et al., 2021). GOHC estimates based on Argo show also smaller uncertainty in general. However,

although GOHC estimates based on Argo are built from the same temperature and salinity Argo profiles, they show some differences that are due to the processing (e.g. selection of valid profiles, gridding algorithm, etc…). Note that the area considered for the Argo-based GOHC change calculation corresponds to the Argo mask, defined in Table 1 and covering 79% of the ocean surface while the geodetic approach is using the altimetry mask that covers 87% of the ocean.

Figure 3 shows the temporal variations of the EEI derived from the LEGOS-Magellium space geodetic dataset as well as that

obtained from the GCOS yearly ensemble and the direct EEI measurements provided by CERES. The 3 solutions detect a trend in EEI over their respective period: 0.29 [0.04;0.56] W m$^{-2}$ decade$^{-1}$ for LEGOS-Magellium over 1993-2022; 0.16 W m$^{-2}$ decade$^{-1}$ [-0.19;0.51] for GCOS over 1993-2020; 0.46 [0.34; 0.59] W m$^{-2}$ decade$^{-1}$ for CERES over 2000-2022. When considering the common 2000-2020 period, LEGOS and Magellium dataset shows a positive trend of 0.39 W m$^{-2}$ decade$^{-1}$ that is closer to the 0.49 W m$^{-2}$ decade$^{-1}$ trend of CERES over the same period. Given the confidence intervals and good

agreement between these independent datasets, these results provide confidence in the observed trend in EEI since 2000, indicating a very likely acceleration in global ocean warming over the periods specified. The Taylor diagram in Figure 4 indicates the similarity in terms of temporal variability of all EEI products with the CERES reference. The proximity of a dataset to the blue star determines the degree of agreement and how well the dataset matches CERES estimate of the EEI variability. The GCOS and LEGOS-Magellium products show close time variations with a correlation of approximately 0.7.



The ARMOR-3D product has the highest correlation (0.84) but also a significant standard deviation. The Argo-based products range from 0.22 to 0.79 in correlation, indicating varying levels of agreement with CERES.

## 4 Discussions and conclusions

In this study we propose an extended estimate of the GOHC change and the EEI from 1993 onwards based on the geodetic approach and we compare it with CERES EBAF estimate of the EEI and from various estimates based on Argo in situ
measurements.

The major advantage of the space geodetic approach is to take into account the whole water column, thanks to the integrated observations of space gravimetry and altimetry since 2002. Comparing space geodetic GOHC with other data sets, mainly based on in situ temperature and salinity profile data of the Argo network, has allowed us to cross-check the consistency of the different estimates. Over the period 1993-2022, the spatial geodetic GOHC shows a significant trend of +0.75 [0.61;1.04]
W m$^{-2}$. Over 2005-2019 the geodetic estimate of GOHC trend is slightly higher than Argo based estimates at the 66% confidence level but it is in general agreement at the 90% confidence level. Besides the difference in spatial coverage of the input data, the discrepancy observed at the 66% confidence level could reveal limitations in the observing systems such as the unobserved deep ocean with in situ data or systematic errors in spatial geodetic data, which need to be further investigated. In addition, the comparison of the geodetic EEI estimate with the direct EEI estimates provided by the CERES
EBAF dataset provides complementary assessment information on the variability of EEI. On the one hand we find a good temporal correlation of the EEI derived from space geodetic and CERES EBAF estimate. On the other hand a significant EEI trend has been detected in both CERES and the geodetic approach suggesting a very likely acceleration of current global ocean warming. This study also highlights the rigorous estimation of uncertainties and their propagation from space geodetic data, based on a mature and advanced state of knowledge of altimetric and gravimetric measurements.

**Data availability**

Space geodetic GOHC change and EEI dataset (v5.0) is available online at https://doi.org/10.24400/527896/a01-2020.003 (Magellium/LEGOS, 2020) with the complete associated documentation (product user manual and algorithm theoretical basis document).

**Competing interests**

The contact author has declared that none of the authors has any competing interests.



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







**Table 1: Impact of the depth and the geographical mask considered in the global integrated expansion efficiency of heat (IEEH) coefficient derived from Argo and ECCO data (Argo mask refers to the most restrictive Argo geographical mask among Argo products see Fig. 1 in Marti et al. (2022)).**

|  | Value of the IEEH coefficient at global scale over the 2005-2015 period (unit: m YJ$^{-1}$) | |
| --- | --- | --- |
| Geographical area and depth | **Argo** | **ECCO** |
| **Argo mask, 2000m** | 0.145 | 0.145 |
| **Argo mask, 6000m** | 0.167 | 0.168 |
| **Extension near coasts, 6000m** | Not available | 0.150 |

**Table 2: Data used to calculate the space geodetic ocean heat content change and Earth energy imbalance and to perform comparisons.**

| Product ref No | Product ID & type | Data access | Reference |
| --- | --- | --- | --- |
| 1 | Sea level gridded data from satellite observations for the global ocean from 1993 to present. | EU Copernicus Climate Change Service, (2018) | Dataset : Lopez, 2018 Publication: Legeais et al. (2021) |
| 2 | LEGOS gravimetric (GRACE, GRACE-FO) ensemble of manometric sea level solutions. | LEGOS FTP site: http://ftp.legos.obs-mip.fr/ pub/soa/gravimetrie/grace_ legos/V1.6/ | Update of Blazquez et al., (2018) |
| 3 | Estimating the Circulation and the Climate of the | NASA ECCO-group website | Dataset: ECCO Consortium et al., 2023. |



| | Ocean - Central Production Version 4 Release 4 (ECCOv4r4) | | Publication: Forget et al., 2015; Consortium et al., 2021. |
|---|---|---|---|
| 4 | Mass contributions to global mean sea level - data set of the European Space Agency Sea Level Budget Closure Climate Change Initiative (SLBC_cci) | CEDA archive | Dataset: Horwath et al., 2021. Publication: Horwath et al., 2022 |
| 5 | LEGOS-Magellium GOHC change/EEI dataset, v5.0 | CNES AVISO website | Dataset: Magellium/LEGOS, 2020 Algorithm Theoretical Basis Publication: update of |
| 6 | GLOBAL_OMI_OHC_area_averaged_anomalies_0_2000; Numerical models, In-situ observations, Satellite observations | EU Copernicus Marine Service Product, 2021. | Quality Information Document (QUID): von Schuckmann et al., 2021. Product User Manual (PUM): Monier et al., 2021 |
| 7 | GCOS EHI Experiment 1960-2020 | World Data Center for Climate at DKRZ | Dataset: von Schuckmann et al., 2022. Publication: von Schuckmann et al., 2023. |
| 8 | ISAS20 temperature and salinity gridded fields | SEANOE - Sea Scientific Open Data Publication | Dataset: Kolodziejczyk et al., 2021 Publication: Gaillard et al., 2016 |
| 9 | Scripps institution of oceanography (SIO) - Roemmich-Gilson Argo Climatology | UCSD SIO Argo website: https://sio-argo.ucsd.edu/RG_Climatology.html | Publication: Roemmich and Gilson, 2009 |
| 10 | Met Office Hadley Centre observations datasets: EN4.2.2. (c14) | MetOffice website: https://www.metoffice.gov.uk/hadobs/en4/download-en4-2-2.html | Publications: Good et al., 2013; Cheng et al., 2014; Gouretski and Cheng, 2020. |
| 11 | JAMSTEC Argo product - | JAMSTEC website : | Publication: Hosoda et al., 2010 |



| | Grid Point Value of the Monthly Objective Analysis using the Argo data (MOAA GPV), version 2021 | https://www.jamstec.go.jp/argo_research/dataset/moaagpv/moaa_en.html | |
|---|---|---|---|
| 12 | NOAA (National Oceanic and Atmospheric Administration) - NCEI (National Centers for Environmental Information) product | NCEI-NOAA website : https://www.ncei.noaa.gov/access/global-ocean-heat-content/ | Publication: Levitus et al., 2012; Garcia et al., 2019 |
| 13 | CERES Energy Balanced and Filled (EBAF) TOA and Surface Monthly means data in netCDF Edition 4.2. | NASA Atmospheric Science Data Center | Dataset: DOELLING, 2023 Publications: Loeb et al., 2018; Kato et al., 2018. |

**Figure 1: Global ocean heat content change over 1993-2022 depicted by the LEGOS-Magellium space geodetic dataset (red curve)**
**and the GCOS dataset available until 2020 (purple curve). The LEGOS-Magellium dataset is characterised by its standard uncertainty envelope [16-84%]. Trends are estimated over 1993-2020 at 5-95% confidence interval level.**



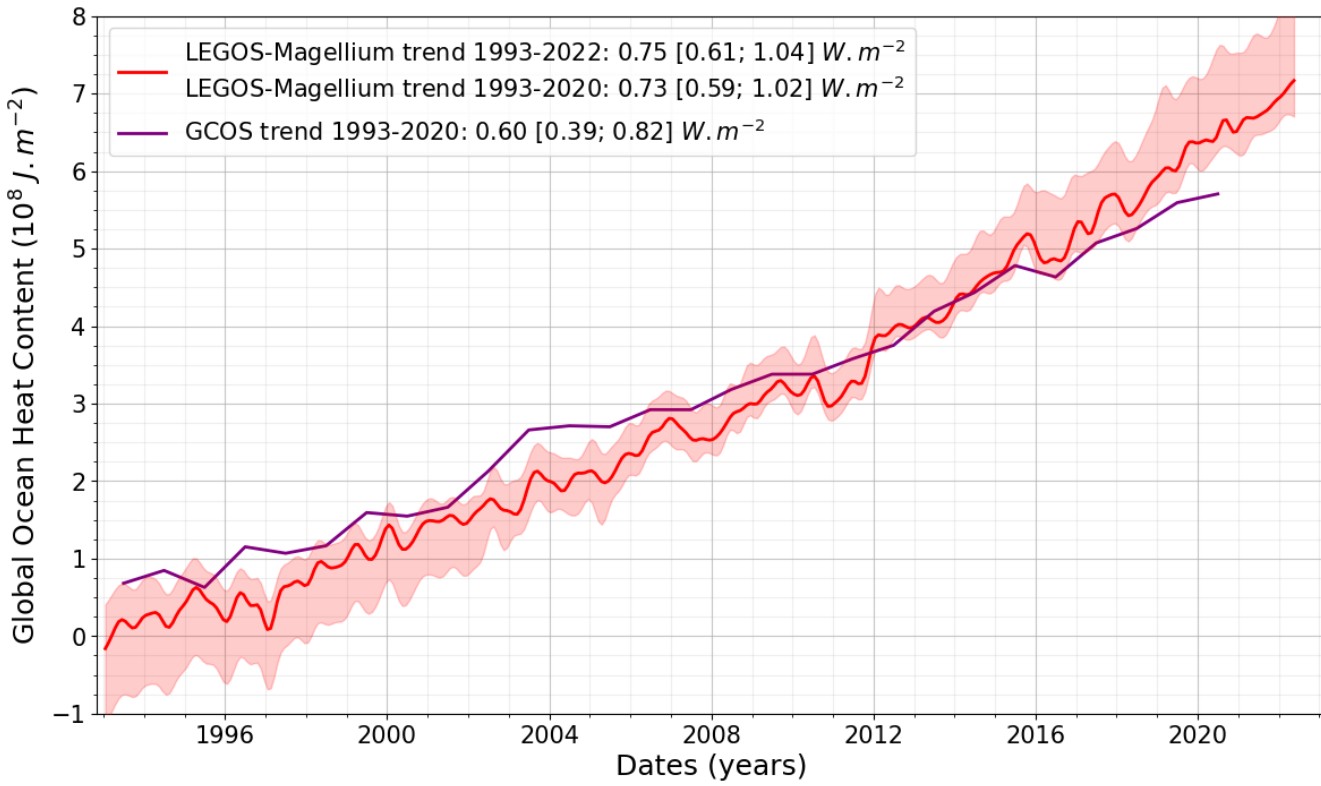

**Figure 2: Global ocean heat content (GOHC) trends over the period 2005-2019 from the LEGOS-Magellium space geodetic dataset**
**(red), the GCOS ensemble (purple), Argo-based GOHC change time series (brown tones), and the 3 CMEMS indicators (green/blue tones). The indicated trend intervals correspond to the 5-95% confidence interval level. ISAS20, SIO, EN4.c14, JAMSTEC, NOAA, CORA and ARMOR3D GOHC trend uncertainties correspond to the adjustment error by the ordinary least squares method.**





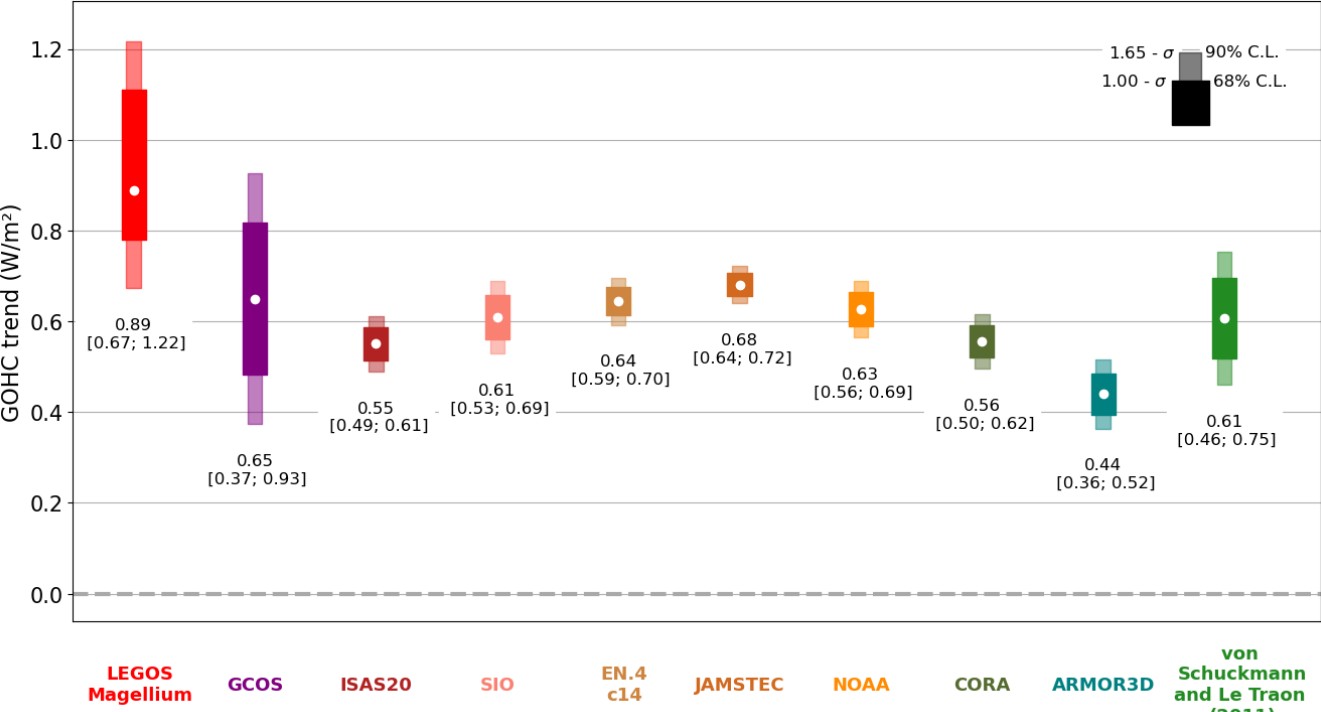

**Figure 3: Earth energy imbalance (EEI) time series derived from the LEGOS-Magellium space geodetic approach (black curve), GCOS dataset (purple curve) and from satellite CERES measurements (blue curve) over 1993-2022. A 3-year filter is applied to the space geodetic GOHC before derivation into EEI. CERES time series is also filtered at 3 years for comparison. Standard uncertainty envelope [16-% 84%] is shown for the space geodetic dataset in grey. EEI trends are given for each dataset on their availability period and uncertainties are estimated at 5-95% confidence interval level.**





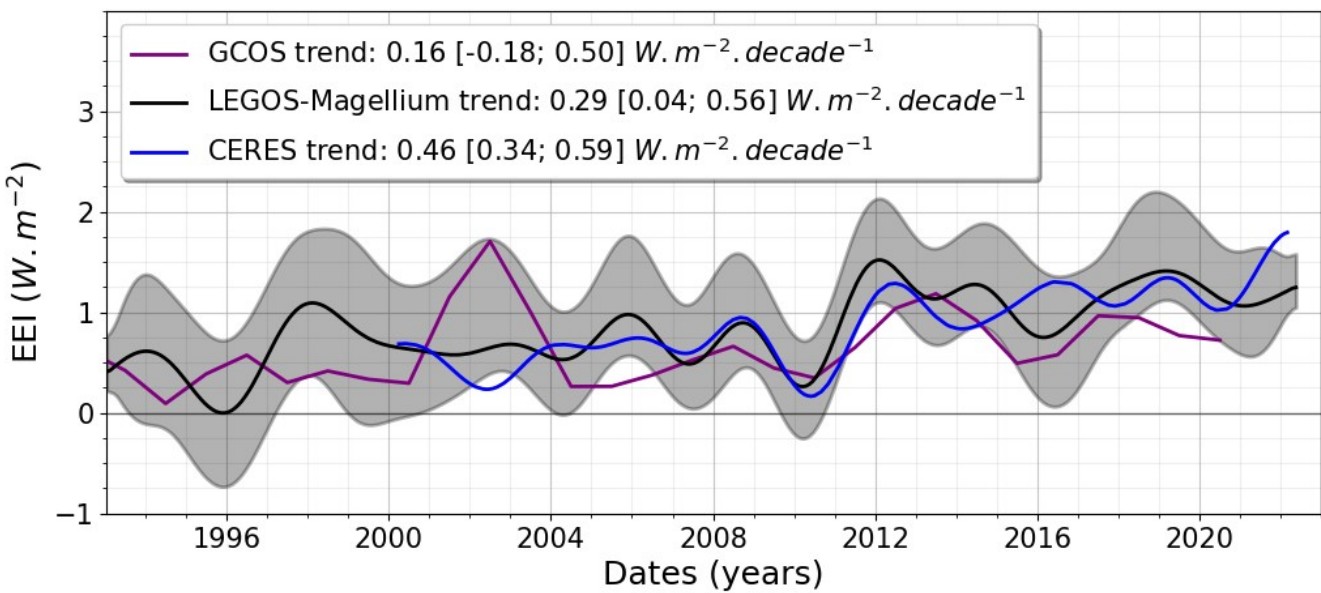

400

405

410 **Figure 4: Comparison of Earth energy imbalance (EEI) interannual variations with respect to the CERES dataset (blue star) on the 2005-2019 period. Taylor diagram gathering the correlation Pearson coefficient, the centred root means square (W m⁻²) and the standard deviation (W m⁻²) for the LEGOS-Magellium dataset (red), the GCOS dataset (purple), the Argo-based EEI time series (brown tones), and the CMEMS indicators (green/blue tones).**





