# Peer review of "Monitoring global ocean heat content from space geodetic"

_State of the Planet, 2023_

## Author Comment (AC1)

**Review Referee #1**

First of all, we would like to thank the reviewer for its relevant questions and comments that will help improve the manuscript, both in terms of content and form.

This manuscript presents analyses of a time-series of EEI using satellite and in situ data, and compares it to time-series from Satellite (CERES) and in situ only estimates. One issue with this manuscript is that the estimate is framed as being "from space geodetic observations" but the key IEEH parameter is estimated from in situ data of ocean warming. Without those in situ measurement of ocean warming, the method would not be feasible. This fact needs to be emphasized in the revised manuscript.

The space geodetic GOHC/EEI estimates are indeed not fully independent from in situ observations, however we opted to call the approach "space geodetic because ECCO data (here, or in situ observations in Marti et al., 2022) are only used to derive static information. The mean IEEH value (over 2005-2015) is assumed to be representative of the total study period (see also answer to question#6). In this study, we relied on the ECCO outputs to estimate the IEEH parameter, unlike Marti et al., 2022 for which IEEH was directly derived from in situ observations. We are aware that the ECCO model assimilates Argo and other observational datasets, here is why we chose to illustrate the consistency between the results at global scale within Table 1.

→ We will emphasise in the manuscript that the knowledge of the warming pattern relies on situ observations.

Also, in general, the in situ estimates are much more certain than the "space geodetic" estimate, which should be acknowledged more directly in revision.

In our approach we intended to characterise the uncertainties associated with the input data as well as possible and then propagate them to the GOHC/EEI. Details are provided in Marti et al., 2022. The obtained trend uncertainty range results of the current knowledge on the uncertainties.
The comparison of the GOHC trends obtained with our approach and from in situ observations (Figure 2) clearly illustrates that the uncertainties associated with in situ based GOHC trends are lower than those from the space geodetic approach. However, for SIO, EN4, JAMSTEC, ISAS20, NOAA, the GOHC were estimated from T/S gridded fields. In such a way, we are able to generate GOHC timeseries in a homogeneous way, by considering data over a same mask (the one described as in Von Schuckmann et al., 2023, hereafter called "GCOS mask") for instance. On the other hand, no uncertainty associated with T/S data was accounted for. The uncertainty trend eventually corresponds to the formal error adjustment of a linear model with the OLS method . No time-correlated effects are taken into account, leading to a lower estimate of the realistic uncertainty. For CORA dataset (and ARMOR3D, but in the next version of the manuscript we decided to simplify it and remove ARMOR3D dataset as it results from a mixed method), we directly used the GOHC time series which is not characterised by any uncertainty. Here also, the uncertainty is given by the formal error adjustment of a linear model with the OLS method.
To sum up, most of the in situ derived GOHC trend uncertainties are underestimated as they omit potential time-correlated errors. Moreover, the uncertainty linked to deep

ocean contribution (we used here the Purkey and Jonhson estimate) is not taken into account. They are therefore not comparable to the other uncertainties. Consequently, the reduced uncertainty ranges do not mean in practice that the in-situ GOHC trends are more reliable than space geodetic GOHC trends. Work is currently on-going (see presentation by Cheng et al. "An update for IAP ocean heat content data and implications for EEI" in 2023) to provide a new estimate of the GOHC accounting for realistic uncertainties related to the measurement, the different biases, the climatology, the interpolation, and mapping method. See also answer to question 3.

→ we will modify Figure 2 (preview of next version of Figure 2 below) so that the results whose trend uncertainty corresponds to the formal OLS error without characterisation of the data uncertainties are not interpreted by the reader in the same way as other results.

[Figure]

Figure R1-1: next version of the Figure 2 of the manuscript

Given these increased uncertainties, more mention of what the "space based" method might add to the in situ method in the revision might also be helpful.

→ In the data and method section, we will add a short sentence to stress on the fact that the space geodetic approach allows accessing information related to the whole water column.

→ We will clarify the different masks used in the paper. First, we will no longer refer to the "Argo mask" because it is confusing for the reader. Figure 2 will be divided into 2 parts as shown on the figure above. On the left, GOHC is estimated considering 86% of the ocean surface (equivalent to altimetry data availability), while for results on the right data was considered over the GCOS mask. Impact of the surface on the GOHC trend is strong and thus with this figure we will highlight one of the assets of the space geodetic approach, that is to say the large spatial coverage of the input data.

Comments follow indexed by line number (L) where possible.

1. L9-10. The word "trend" is not a correct choice. Consider changing to "mean value".

2. L10. The word "indicating" is not correct. The mean value does not indicate an acceleration. Perhaps change to "with an estimated acceleration of 0.29 W m-2 decade-1".

→ This will be corrected in the new version of the manuscript. We want to talk about EEI

trend, and therefore acceleration in heat accumulation.

3. L23-29. In situ estimates relying mostly on Argo data are already there in terms of accuracy of 0.1 W m-2 on a decadal time-scale, at least starting from circa 2006 when Argo first achieved sparse near-global coverage. This fact should be indicated in the revised text.

As mentioned above, in the current manuscript the uncertainties associated with in situ based GOHC trends are indeed about 0.1 W/m² (at 90% confidence level) over 2005-2019. They are underestimated because they only correspond to the formal error adjustment of a linear model with the OLS method, where the potential time-correlated errors are not taken into account.
The paper from Von Schuckmann et al., 2023 illustrates that the trend uncertainty associated with in situ based GOHC data (here "GCOS") resulting from a community effort is comparable with the uncertainty range obtained with the space geodetic method (0.68 ± 0.3 W/m² over 2006-2020, full depth, 95% confidence level).
Cheng et al. "An update for IAP ocean heat content data and implications for EEI" showed early 2023 the first results of an ensemble approach for estimating uncertainty associated with GOHC (0-2000m) from in situ measurements. In slide 23, without uncertainty related to the mapping, they obtain σ: ~0.15 W/m² over 2000-2022, that is to say 0.25 W/m² (90% CL) over 20 years without considering the full water column.
These various elements do not allow us to conclude that the accuracy objective for the EEI study has been achieved with in situ measurements.

4. L51-56. The Roemmich and Gilson temperature and salinity maps include mitigation against the influence of salty drifters that appear to be largely effective at removing the halosteric bias error.

From Roemmich and Gilson we understand that this is the dataset from the SIO. Indeed, the SIO data could be used to remove the halosteric component, however these are only available from 2004 and cannot be used for our study which extends back in time to 1993.
→ eventually we might opt to remove the sentences related to recent instrumental drifts because it brings more confusion than clarifications to the reader.

6. L57-64. This is really the heart of the method, and reliance on Argo data to estimate the warming pattern and hence IEEH should be explicitly noted. Without in situ data to show where (at what temperatures, pressures, and salinities) the warming is taking place, the method uncertainties would be unacceptably large. So this is not really monitoring ocean heat content changes from space as the title implies! It requires monitoring the changes in situ to work with any accuracy. Also, ECCO is very poor at reproducing observed warming in the deep and bottom waters. It shows global cooling from 1992-2011 (e.g., Wunsch and Heimbach, 2014, J. Phys. Oceanogr.) whereas repeat hydrographic data (e.g. Purkey and Johnson, 2010) and Deep Argo float data show very definite warming in the global mean.

→ In the section Data and method, we will mention that the warming pattern is assessed from in situ observations.

On the time scale characteristic of the study, we do not expect any temporal variations associated with the warming pattern. This is already known at global scale (Kuhlbrodt and Gregory, 2012) and it can also be illustrated at regional scales with an Argo based product (see figure below) that the order of magnitude of the time variations associated to the regional IEEH are <4% in comparison with its mean value. This justifies why we used the IEEH temporal mean value, computed over 2005-2015 (Argo golden area), for our study over 1993-2022.
→ We will specify in the text the period we considered for the mean IEEH.

[Figure]

Figure R1-2 : temporal variations of the IEEH estimated from ISAS20 gridded T/S data over 2002-2020 (0-2000m) – ratio (%) IEEH trend over IEEH mean

Moreover, we found out that the choice of the in situ dataset has a very low influence on the IEEH value. In Marti et al, 11 in situ T/S solutions over 0-2000m were used to compute the global IEEH and it resulted in 0.145±0.001 m/YJ (90% CL), ie less than 0.7% discrepancy between global IEEH mean values.

In this study, we used ECCO to estimate the IEEH. ECCO is indeed subject to model drifts and bias of other observational systems (omissions errors), however consistent results have been obtained in the comparison of the IEEH means from the ECCO model and from in situ data. We assessed the discrepancies between the mean IEEH values over their common geographical area. Results at global scale are given in Table 1 (in the next version of the manuscript, global IEEHs in Table 1 will be estimated over the GCOS area and no longer on the Argo mask for clarity reasons). Results at regional scales are illustrated on the figure R1-3: we estimated the differences between regional IEEH derived from ECCO data and regional IEEH derived from T/S gridded fields based on in situ observations and provided by 2 centres (combination of ISAS20 and EN4.2.2.l09 data). Regional IEEH values correspond to the mean value (0-6000m) over the period April 2002-December 2020. The differences do not exceed 5% in the open ocean.

[Figure]

Figure R1-3: Comparison between regional mean IEEH, from ECCO and from in situ (ISAS20, EN4.2.2.l09) gridded T/S (0-6000m). Mean values are estimated over April 2002-December 2020.

7. L65 & L68. Superscript formatting was lost in the exponential numbers. Corrected;

8. L85-88. This portion of the manuscript requires a bit more exposition. Unless the filtering is strongly non-linear, it should not matter where it is done filtered regionally and then summed, or summed and then filtered.

We agree that filtering before or after summing should not impact. We however observed an impact on the EEI, the Lanczos filter that we are using is not perfectly linear (two lobes). There is no impact on the GOHC timeseries but when we differentiate the effect of this non-linearity is accentuated.

→ For ease of reading we will simplify this portion.

9. L89-99. Is the uncertainty of the IEEE coefficient included in the overall estimate? If not, please justify why that can be neglected. If it is, please describe that portion of the estimate.

The uncertainty associated with the IEEH coefficient was also used and propagated until the GOHC and EEI using error covariance matrices. The contribution of this uncertainty to the EEI mean uncertainty is negligible and represents <0.1% in variance. An additional study has shown that 75% of the EEI mean uncertainties (variance) are coming from uncertainties associated with the gravimetry data while the remaining part is due to altimetry sea level data.

→ in the text portion that describes the error covariance matrices computation, we will mention that the uncertainty associated with the IEEH had been propagated and eventually negligible compared to other uncertainties.

10. L95. Generally phrases like "Note that" or "Is is notable that" are superfluous and can be deleted. If it's not noteworthy, why put it in a manuscript? Deleted.

11. L116-118. Hasn't the geodetic series been filtered with a 3-year low-pass, so really the comparisons are only for signals with periods of about 3 years or greater?

→ " both GCOS ensemble and OMIs are made up of yearly .. so that the derivative is made on a monthly time scale": this portion needs to be clarified. We will explain in this portion that GOHC annual time series are used. They are then linearly interpolated to monthly time steps before applying the same method as applied for our product to compute the EEI. It will therefore mean that a 3 year low pass filter was applied to all time series.

12. L121-123. Please mention here that the CERES EBAF product mean EEI has been anchored with in situ estimate (including an ocean estimate that relies primarily on Argo data).

Thanks, you are right, it is important to mention it for the Figure 3.

→ We will add in the next version of the manuscript that CERES EBAF product mean EEI has been anchored with in situ product from Lyman and Johnson (2014).

13. L149. Generally sentences like "Figure x shows" are poor topic sentences, and duplicate figure caption. How about "Temporal variations of EEI derived from ....agree fairly well with those obtained from the GCOS yearly ensemble and CERES measurements (Figure 3)." Reformulated.

14. L150-161. Loeb et al. (2021) showed a pretty good agreement between an in situ estimate and CERES both for the acceleration of EEI, and the interannual correlation. Their correlation at 1-year resolution is comparable to those here at 3-year resolution.

→Thanks. Loeb et al., 2021 will be mentioned.

15. Figure 3 and discussion. It would probably be more interesting to the reader to compare trends over the portion of the record common to all three time-periods.

EEI trends over the common period 2000-2022 are given in lines 153-154, while EEI trends over individual availability periods are shown on Figure 3.

→ For the sake of clarity, we will review Figure 3 to display trends over the common period 2002-2022.

---

## Author Comment (AC2)

**Review Referee #2**

First of all, we would like to thank the reviewer for its relevant questions and comments that will help improve the manuscript, both in terms of content and form.

Marti et al. provide an update on the global ocean heat content change since 1993 based on a geodetic approach. The new time series shows a positive trend of OHC-based EEI, suggesting an acceleration of global heating. This geodetic-approach-based estimate shows a reasonably good agreement with other in situ observations. The paper is good and well-written, and the geodetic approach provides a promising method to estimate OHC and EEI, which should appear in the state of the ocean report. I have several comments on a couple of things and hope to improve this paper:

Abstract: a trend of 0.75 Wm-2 does not suggest an acceleration of ocean warming, instead, it is "ocean warming". The number you derived later: 0.29 [0.04;0.56] W m-2 decade-1 for LEGOS-Magellium over 1993-2022 is the "acceleration". please rewrite the abstract.

In the abstract we want to focus on the acceleration of the EEI, not the global OHC one.

→We will correct the abstract to display the warming acceleration (ie EEI trend).

Abstract: The last sentence "This study highlights the importance of rigorously estimating uncertainties based on space geodetic data to robustly assess EEI changes.". This seems not a key point of this study. The uncertainty quantification is always referring to other papers and not thoroughly introduced in this paper. I would say something to highlight the importance of a combination of multiple lines of evidence to depict ocean warming and its acceleration, because clearly, the geodetic approach shows the value and could be a nice addition to the traditional approaches.

→ the end of the abstract will be reviewed to focus on the importance of analysing various estimates and their uncertainties, and of the added value of the space geodetic method, to eventually robustly assess the EEI changes.

Introduction, page-1, line 20-24: please give a balanced and more inclusive citation for each group of methods.

For the in situ approach, we eventually only kept von Schuckmann et al., 2022 as it is the latest publication regarding this topic that aims at presenting results from the scientific community. Stammer et al., 2016 provides a review of data assimilation issues and challenges for ocean reanalysis for climate applications and seems relevant enough to illustrate the method.

→ Hakuba et al, 2021 will be added for the space geodetic approach.

Page-2, line 48-50: this paragraph seems out of place, I suggest putting it after page-3, line 74, i.e. the derivation of GOHC should come after the regional OHC is derived. And, another thing is: please also clarify how you make regional patterns before GRACE (you used the individual contributions to manometric sea level from Greenland, Antarctica, mountain glaciers and from terrestrial water storage before 2002, are you using regional fingerprints or just global time series?)

- The details given on line 48-50 were located here to highlight the difference in the method compared to Marti et al., 2022 based on the progress made in Rousseau et

al., under review, and to show that GOHC change is no longer estimated directly from global steric level change as in Marti et al. 2022, but as the sum of regional OHC changes. In this manner, it helped us to explain the calculation of variables (SSL, IEEH) at regional scales.

→ so as not to interrupt the thread in the method between the description of the input data and the calculation of the SSL, we propose to slightly reword the lines 48-50 and move them higher up, before the description of the input data.

- Regarding the barystatic component of sea level beyond the GRACE(-FO) period, we relied on the work performed in the frame of the ESA-CCI program about the sea level budget closure (Horwath et al. 2022). It is estimated by summing several contributions:
    - The contribution from global glacier mass changes assessed by a global glacier model
    - The contribution from Greenland Ice Sheet mass changes assessed by satellite radar altimetry and by GRACE
    - The contribution from Antarctic Ice Sheet mass changes assessed by satellite radar altimetry and by GRACE
    - The contribution from terrestrial water storage anomalies assessed by the global hydrological model WaterGAP (Water Global Assessment and Prognosis)

Time series is provided at global scale, and it is then considered homogeneous over the global ocean. This approximation is motivated by the fact that the contribution of the mass to spatial variations of sea level change is only significant in closed seas and very high latitudes (Piecuch and Ponte, 2011; Piecuch et al., 2013) which are not present in our study. Moreover a sensitivity test was conducted over the GRACE era to assess the impact on the GOHC and EEI of considering homogeneous manometric sea level maps instead of regional patterns. The results showed a difference of around 1% in the GOHC trend and no more than 3% difference in the interannual variations of the EEI.

→ we will specify in the manuscript that the SLBC data are defined at global scale.

Page-2, line 51: Is the salinity effect only neglected in GOHC calculation, not for regional OHC calculation?

We neglected the salinity effect in the OHC, and therefore in the GOHC calculation, due to the lack of in situ based halosteric sea level change data which is both available over the study period and corrected for the drift due to recent anomalies on conductivity sensors. Moreover, the study concentrates on the OHC on a global scale, to which the halosteric contribution to sea level is negligible compared with the 2 other contributions.

→ eventually we might opt to remove the sentences related to recent instrumental drifts because it brings more confusion than clarifications to the reader.

Page-3, line 85-86: I don't understand why performing the filter step at regional scales is low on GOHC estimate but more on the EEI estimate. What does exactly this mean? Isn't EEI is simply derived from GOHC according to Eq.(1)?

The Lanczos filter that we are using is not perfectly linear (two lobes). The effect of this non-linearity is accentuated when we differentiate the GOHC into EEI.

→ For ease of reading we will reformulate to simplify this portion related to the filtering step.

Page-4, line 100-119: the presentation and introduction of these datasets are a bit of chaos and not organized: different data products should be better grouped according to the data/methodology difference, so the readers can better understand their differences. Please see Cheng et al. 2022 (https://doi.org/10.1038/s43017-022-00345-1 ) for reference.

Page-4, line 111-115: Not all five data centers are "Argo centers". They are fundamentally different from "Argo resulting GOHC change estimates" because ISAS, EN4 and NOAA used all available in situ data, but JAMSTEC and SIO are purely Argo. They should be better grouped and introduced.

→ The data for comparisons will be better described. We will first introduce GOHC computed by ourselves from gridded fields based on in situ data, starting with Argo-based ones. We will then introduce the 2 OMI CORA and CORA-2011 also based on in situ data (in the next version of the manuscript, we decided to simplify it and remove ARMOR3D dataset as it results from a mixed method). GCOS ensemble, gathering most of those datasets. Finally, we will introduce another space geodetic GOHC (Hakuba et al., 2021) .

When they are not appropriate, references to "Argo-based data" in the section 2 (IEEH description) and in the section 3-Results will also be removed.

Page-4, line 111-115: all five data products are used in the GCOS estimate, so they are not independent from the GCOS estimate. Please discuss this issue.

→ This information and associated comment will be given (currently only CORA and CORA2011 inclusion within the GCOS ensemble is mentioned on line 110-111).

Page-4, line 119-121: I don't understand this, many of the data products listed above are monthly data (ISAS, EN4, NOAA, JAMSTEC, SIO, CORA, ARMOR-3D), why should you need to interpolate to monthly time series??

Page-4, lines 116-117 indicate that we chose to subsample all the GOHC monthly datasets to fit with the annual time-series from GCOS and CMEMS-OMIs (particularly relevant for this publication in the OSR).

→ " both GCOS ensemble and OMIs are made up of yearly .. so that the derivative is made on a monthly time scale": this portion needs to be clarified. We will explain that GOHC annual time series are used. They are then linearly interpolated to monthly time steps before applying the same method as applied for our product to compute the EEI.

Page-5, line 136: "Over their respective area of data availability": please clarify which domain you are referring to?

By "their respective area of data availability", we meant that the LEGOS-Magellium GOHC was computed considering the maximum spatial extent our method can reach at the moment (~86%), while for GCOS the dataset has its own and fixed spatial extent reference (~76%) - for GCOS, we do not have control on the spatial extent, the GOHC change time series associated with von Schuckman et al., 2023 publication was used)

→ In order to facilitate the comparison between GCOS and Legos-Magellium GOHC trends in Figure 1, we will update Figure 1 to plot the LEGOS-Magellium GOHC change obtained over the same spatial extent as the GCOS dataset. We will also revise the comment.

→ we will update Figure 2 and divide it into 2 parts as shown on the figure below. On the left, GOHC is estimated considering 86% of the ocean surface (the maximum spatial extent that can be taken into account with our approach currently), while for results on the right data was considered over the GCOS mask.

[Figure]

Figure Ré-1: next version of the Figure 2 of the manuscript

Page-5, line 146-148: it is not correct: different GOHC estimates used in this study has different ocean-land mask, so their areas are different. And there is no "Argo mask" because different Argo data center also has different masks. Please clarify what is this referring to.

We called "Argo mask", the mask defined in Marti et al. 2022 (figure 1). It corresponds to the common availability area of 11 in situ gridded T/S datasets and covers 79% of the total ocean surface.

Apart from GOHC changes estimates on which we do not have control (OMIs, GCOS), we can ensure that the area considered to compute the different GOHC change estimates is homogeneous using a common mask. Even if the ocean-land border used by the different in situ data centers are different, in such a way we are able to discard the main discrepancies on GOHC estimates related to spatial extent of their input data.

→ In the next version of the manuscript we will no longer refer to the Argo mask when discussing the different GOHC trends because it brings confusion. Only two masks will be mentioned: see figure above R2-1.

Page-5, line 149-154: please also discuss the variability in these time series: are the inter-annual variation meaningful (i.e. ENSO) or just noises?

Some signals can result from errors, but most of them are eventually not noise as there is correlation between them. These variations can be related to the main ocean modes (PDO, ENSO) or the interannual variability of the aerosol content in the atmosphere (volcanic eruptions), and are not well known and documented.

→ We will add a sentence in the next manuscript to discuss this aspect.

Table 1: I don't know what is "Argo mask" you are referring to: mask of which Argo product?

See above, "Argo mask" is an internal definition that we will no longer use

---

## Author Response (AR2)

**Minor comment from editor:** : What is meant in line 57/58 with " It now includes coastal regions up to 100km" - is that the distance from the coastline?

Changed: "It enables the inclusion of coastal regions up to 100 km **from the coastline** and deep ocean areas down to 6000 m."